# New Insights about Chronic Pelvic Pain Syndrome (CPPS)

**DOI:** 10.3390/ijerph17093005

**Published:** 2020-04-26

**Authors:** Keren Grinberg, Yael Sela, Rachel Nissanholtz-Gannot

**Affiliations:** 1Department of Nursing, Faculty of Social and Community Science, Ruppin Academic Center, 40250 Emek-Hefer, Israel; 2Department of Health Systems Management, Ariel University, 40700 Ariel, Israel

**Keywords:** chronic pelvic pain syndrome (cpps), therapeutic interventions, musculoskeletal pain, urology, gynecology

## Abstract

Background: Chronic pelvic pain syndrome (CPPS) is one of the common diseases in urology and gynecology. CPPS is a multifactorial disorder where pain may originate in any of the urogynecological, gastrointestinal, pelvic musculoskeletal, or nervous systems. The symptoms of CPPS appear to result from an interplay between psychological factors and dysfunction in the immune, neurological, and endocrine systems. The aim of this article was to present new insight about CPPS in order to raise awareness of nursing and medical staff in the identification and diagnosis of the syndrome and to promote an appropriate treatment for each woman who suffers from CPPS. Methods: A literature review about the factors associated with CPPS and therapeutic interventions for CPPS was conducted. Results: CPPS represents a chronic pain syndrome that combines anatomic malfunction of the pelvic floor muscles with malfunction of pain perception linked with psychological and cognitive factors. Conclusions: The therapeutic interventions in CPPS cases should, consequently, follow a multidisciplinary approach.

## 1. Overview on Women with CPPS 

Chronic pelvic pain syndrome (CPPS) is defined as pain located in the pelvic area that lasts over six months and is severe enough to limit functioning, unrelated to menstrual cycle, pregnancy, local trauma, or pelvic operations. This syndrome is one of the diseases shared by urology and gynecology [1,2,3]. Its frequency is between 3% and 10%, and it is more frequent among women [4,5,6]. The costs of treating CPPS were estimated at about USD 880 million annually [7]. About 15% of women reported loss of work days and 45% reported decreased work efficiency [5,8].

This integrative review was aimed at presenting an interdisciplinary overview on women CPPS patients. Therefore, a descriptive review of the literature on the pathogenesis, risk factors, diagnosis, and treatment strategies was performed.

## 2. Materials and Methods

### Search Strategy

A systematic literature search, using PubMed, Medline, Embase, PsychINFO, and Web of Science databases, was performed by two authors with the terms and phrases: "Chronic pelvic pain", "painful bladder syndrome", "provoked vestibulodynia", "musculoskeletal pain", and "therapeutic interventions".

Only publications in the most recent decade were searched for, and the date of publication was set from 1st January 2010 to 31st January 2020. A manual search was conducted to identify additional potential eligible studies from the reference lists of the eligible articles. All authors reviewed the search strategy. Overall, we selected 92 publications. In this review, we tried to provide an overview of a large and complex topic to provide indications regarding the management of CPPS patients, since management requires a holistic approach in order to provide patients with proper care. Painful bladder syndrome (PBS) and provoked vestibulodynia (PVD) are subgroups of CPPS [1]. Diagnosis of PBS is based on complaints that include an urgent and/or frequent need to urinate and pain that disrupts daily activities and reduces quality of life (such as interrupted sleep, difficulty with sexual intercourse, depression, or anxiety) [1,9,10,11]. Manual examination of the pelvis may find tension, rigidity, and sensitivity of the pelvic floor muscles or myofascial trigger points. PVD refers to pain at the entrance of the vagina, known as the vestibule, experienced as sharp or burning pain that lasts at least three months [12,13,14]. The diagnosis criteria for PVD may include severe pain when touched or when vaginal penetration is attempted, soreness located in the vestibule without similar soreness in adjacent tissues, and ruling out other factors (infection, inflammation, skin disease, etc.) [13]. PVD is diagnosed with a Q-tip test. A doctor assesses pain by touching the vestibule with a cotton swab [12,14]. Most patients report pain during sexual intercourse to the point that it is impossible, and mention pain during gynecological examinations [15], while inserting a tampon, or with direct contact such as while bicycle riding, horseback riding, or wearing tight clothing [12,14,16,17,18] (Figure 1). 

A number of mechanisms have been suggested as the pathophysiological basis of CPPS: (1) An infection process, but no conclusive evidence has been found and antibiotic treatment is ineffective; (2) A neurogenic inflammation that includes local chemical changes [1,7]; (3) Hypoxia. A disrupted blood flow to the pelvic area, reduced micro-vascular density of the bladder’s submucosa layer, followed by decreased perfusion [19] is supported by clinical improvement following hyperbaric treatment [20,21,22]; and (4) Weakness or cramps of the pelvic floor muscles [13,23]. None of the above have been mentioned as the sole cause of the syndrome, and it is estimated that these mechanisms interact. A common explanation is that, for an unknown reason, the glycosaminoglycan (GAG) layer that coats the mucosa of the bladder and vagina is damaged. This damage leads to a chain of nerve-cell-level processes that culminate in a neurogenic inflammation and mast cell activation [24]. The GAG layer, which is an impermeable barrier of urine solutes, becomes permeable. The infiltration of solutes into the submucosa irritates the nerve endings and creates heightened inflammation agents: vasoactive intestinal peptides, substance P, and acetylcholine [16,24]. As a result, activated mast cells secrete histamines that widen the blood vessels and generate a local inflammation [25]. The inflammation irritates the C-fibers and causes increased release of more inflammation agents, which in turn cause additional damage and the formation of fibrosis on the bladder [21,25] or in the vagina [15,16,24,26]. 

The literature has indicated that PBS and PVD are chronic pain diseases [27], which involve dysfunction of the general pain system, and are not just a pelvic dysfunction. Concerning PBS, research findings demonstrated hypersensitivity in response to experimental pain among these women [28] expressed by allodynia and hyperalgesia in the projected pain area above the pelvis [21,29,30]. Women with PVD, in comparison to healthy women, presented increased sensitivity to pain in body areas distant from the genital area [31,32,33,34,35,36] conveyed by allodynia and hyperalgesia. Harlow and Stewart (2003) [37] found that PVD women demonstrated disruptions of pain modulation as well as increased central sensitization processes, but little is written in the literature about this dimension. So far, only two imaging studies examined central pain processes in PVD women, and both found high activation in response to painful experimental stimulation. One study [38] found increased activation of the several brain areas, amygdala, thalamus, anterior cingulate cortex (ACC), prefrontal cortex (PFC), and insula, in response to pressure stimulation of the vulva. A second study [39] found activation and hyperactivity in the same brain areas, in comparison to healthy subjects, in response to pressure stimulation of a distant area or of the vulvar area. Imaging studies that examined CPPS patients in general reported reduced grey matter in the ACC [40]. The relative decrease of grey matter volume, which indicates neuroanatomic changes, was found in other chronic pain conditions such as chronic back pain, inflammatory bowel syndrome (IBS), migraines, and phantom pains [41,42,43]. Furthermore, it is possible that structural changes are related to depression and other emotional aspects that accompany chronic pain [42,44,45], because areas such as the ACC have a significant role in processing emotional information in general and that related to chronic pain specifically. 

These findings, which indicate hypersensitivity of the pain system among women with PBS and PVD, raise the possibility that a dysfunction of the central pain system is at the core of CPPS. However, it is not clear whether a peripheral defect that causes chronic pain affects pain processing on the level of the central nervous system and creates changes in the perception and processing of pain [46] or, alternatively, a dysfunction of pain processing and modulation causes the development of chronic pain in CPPS women [1,24].

The existence of other, especially idiopathic, chronic pain diseases in CPPS patients (such as fibromyalgia and IBS) supports the notion that a dysfunction of the central pain system is the source of the pain. For example, among patients with medically unexplained symptoms, 19% met diagnostic criteria for both chronic pelvic pain and fibromyalgia [47]. CPPS and fibromyalgia share many unexplained characteristics and key symptoms, including pain as a prominent symptom, comorbidity, and localized and systemic conditions [48]. Women with overlapping pain syndromes may have more widespread symptoms and evidence of central pain sensitization [49]. Studies revealed that women with both fibromyalgia and chronic pelvic pain demonstrate increased widespread pain intensity, anxiety, and depression [50,51]. Presence of pelvic pain is associated with increased overall pain severity and fibromyalgia disease impact. It may be that changes in the effectiveness of pain processing, control, and regulation processes, which characterize women with CPPS, affect the severity of the symptoms and are involved in the response to interventions [1,52]. These variables and their relationship with the severity of CPPS and the effect of the treatment should be examined in future research.Similar to other syndromes in which the pathogenesis process is unknown [53], the treatment of CPPS is varied (Figure 2) and includes:Cognitive behavioral therapy (CBT) includes deep breathing, learning techniques to control (contract and relax) pelvic muscles, and bladder training. Bladder training teaches patients to control the disease (for example, by prolonging periods between urinations by using various mind distracting methods) and has been proven to increase urine volumes and decrease urination frequency [20,22,24];Medications include drugs such as tricyclides and sodium pentosanpolysulfate (which inhibits histamine release from mast cells) or drugs that are injected into the bladder (for example, glycosaminoglycans that are part of the bladder’s natural protection layer; resiniferatoxin and capsaicin that bind to nerve-ending receptors and desensitize pain fibers; and botolonium toxin that inhibits the secretion of inflammatory agents such as substance p and nerve growth factor and prevents higher tonus of pelvic muscles). Most drugs tested on CPPS patients showed good short-term results [30], but only a few were found to have long-term results [20,22]. There is evidence of some success in response to hormone therapy administered orally, by ointment, or by injection [54,55,56]. The various drugs are administered systemically or locally, and they can be classified by their action mechanisms: (1) Improvement of pain regulation (assuming the pain is neuropathic) by drugs that inhibit reuptake of noradrenaline and serotonin, and improve functioning of the downward pain-processing conduits [57,58]; (2) Preparations that delay nerve conduction velocity by using anesthetics; these have been tested by a number of studies, and were found to have limited effectiveness [59]; and (3) Drugs that inhibit production of prostaglandins, or steroid-type drugs, administered due to the assumption that CPPS has an inflammatory source [41] (Table 1);Surgery is a last resort when traditional treatment has failed. Some surgeries of PBS women destroy the bladder nerves and others implant electrodes that electrically stimulate the nerve roots. However, this treatment, as is the case in other chronic diseases, provides only a partial solution, and is not based on empiric guidelines. Therefore, the mechanisms at the root of CPPS must be understood to identify which women are suitable for this kind of treatment [20,60]. Women with PVD are sometimes offered a vestibulectomy, in which the mucous membrane of the vagina, the hymen, and some glands in the area are removed [54,61,62,63]; andMyofascial physical therapy (MPT) is considered safe, is recommended for the syndrome, and is an important stage of therapeutic intervention for the pelvic floor (see the following section) [2,64,65,66,67].

## 3. Myofascial Physical Therapy (MPT)

Physical treatment of the pelvic floor includes, in addition to exercises to strengthen the pelvic floor, biofeedback treatment and electric muscle stimulation, or bladder training. Pelvic myofascial manual therapy (MMT) is performed by an expert physical therapist, and its aim is to release a myofascial constraint at painful trigger points. The treatment is based on making soft tissue more elastic, easing articular rigidity in the area, releasing and stretching shortened muscles, and strengthening weak muscles in order to restore the balance of musculoskeletal components, to allow optimal painless functioning, and to reduce discomfort [68,69]. In addition, this treatment could improve blood flow to the pelvic area; reduced blood flow has been suggested as one of the mechanisms at the basis of the syndrome [30,64,68,70,71,72]. Pelvic floor exercises and bladder training could improve coordination and functioning of pelvic floor muscles [65]. Manual treatment of myofascial trigger points has been proven to reduce pain [55,73,74,75]. Although this treatment is very common and accepted as a method with beneficial results, the mechanisms that occur following this intervention are still unclear, and we still do not have the tools to identify the patients for whom this intervention would be clinically effective [73,74].

The rationale for treating CPPS patients with MMT is based on pathological findings in the pelvic floor area of these patients, which attest to malfunction of the pelvic floor muscles, and include points sensitive to pain as well as additional musculoskeletal abnormalities in the pelvic floor area [68,76,77]. These findings cause pain and other symptoms characteristic of CPPS. Indeed, MMT has been found to reduce pain severity; hence, it would seem that MMT has significant clinical impact among CPPS patients [78]. Additional verification of MPT’s efficacy was found in a study that proved that focused MPT improved the clinical condition and pain of 60% of women with CPPS compared to improvement in 20% of women with CPPS who received general massage [64]. The fact that physical treatment was found to be effective in pelvic pain diseases raises questions about its importance for CPPS patients, which (local or systemic) factors can predict its success, and whether it can affect these factors [78]. 

## 4. Psychological Factors and Pain

The experience of pain and its processing involve many psychological and cognitive variables such as emotions, cognition, focus, a sense of control, adjustment, behavior patterns, interactions between the patient and the therapist, as well as beliefs and expectations [79]. A key variable involved in the experience of pain is pain catastrophizing, which is defined as exaggerated negative orientation toward aversive stimulation. Pain catastrophizing includes three dimensions: rumination on the painful stimulation, magnification of the threat inherent in the painful stimulation, and a self-perception of helplessness to control the pain [80,81]. People with a high degree of pain catastrophizing attribute great severity, threat, or catastrophic consequences to pain [82]. Research has indicated a strong relationship between the pain catastrophizing variable and reports of enhanced clinical and experimental pain, great mental distress, depression, limited functioning, higher post-op reports of pain, lower response to medical treatments, and greater use of medical services [32,52,72,83,84]

Among women suffering from PBS, a relationship was found between the degree of pain catastrophizing and the severity of the pain and number of symptoms [83,84]. Granot and Lavee (2005) [85] noted that whereas no difference in pain catastrophizing in response to pain stimulation to the arm was found between women with and without PVD, the pain catastrophizing scale (PCS) rating of pain during intercourse correlated with higher pain rating among women with PVD. Other studies found that higher pain catastrophizing was reported by women with genital pain in comparison to healthy women [72], and there was a correlation between the rating of pain catastrophizing and increased cerebral activity in the prefrontal brain areas responsible for pain modulation or focus [18]. 

Regarding anxiety, it was found that women with PBS suffered higher anxiety levels (which related to the severity of the symptoms) than did healthy women [83,86]. In addition, the severity of the pain of women with PVD was linked to higher anxiety, beliefs of fear-avoidance, and pain catastrophizing, which indicates that the severity of genital pain could be linked to emotional stress that is characterized by negative expectations such as fear or anxiety [24,52]. Another study reported that apart from the pain problem, additional issues (such as sexual problems) were linked to the anxiety of these women [87].

The fear-avoidance (FA) model was suggested to help understand the psychological processes that occur in response to pain. Accordingly, it can be assumed that among women who suffer from dyspareunia, catastrophic thoughts, fear, and muscle tension form avoidance of painful situations, which creates disability and depression [72]. Anxiety symptoms may reflect an increase in negative feelings, which at later stages (according to the FA model) increase pain catastrophizing [88], and these eventually intensify fear and avoidance of pain [88,89]. It was suggested that anxiety and fear of pain coupled with negative expectations cause the patient to focus on her illness and pain, which affect her sexual response and ability to cope with pain [24]. This supposition was corroborated by a study that found that the perception of pain and anxiety play a role in PVD pathogenesis and contribute to the development of hypersensitivity to pain among these women [85]. Similarly, Heddini and colleagues (2012) [14] found that low levels of anxiety and pain catastrophizing were related to less pain during intercourse and improved sexual functioning.

Another key psychological factor that affects perception of pain is depression. Depression is defined as a complex neurological and cognitive reaction to loss or absence, and as a health condition, detrimentally affects thoughts, emotions, and the ability to function in everyday life. The American Psychiatric Association defined depression as a desperate mood and loss of interest or pleasure in almost all activities that caused pleasure before the event [90,91]. Among women with CPPS, the frequency of depression is higher compared to women who do not suffer from the syndrome [92,93]. Furthermore, among these women, the level of depression correlates with sensitivity to experimental pain and duration of the illness [34]. In light of this, it was suggested that depression has a significant contribution to the severity of CPPS [93].

An additional psychological variable that is common among chronic pain patients is somatization [94]. Somatization is described as a disorder in which the patient complains of a number of somatic symptoms (multisymptomatic) with no organic evidence of pathology. In fact, these symptoms appear as a result of psychological problems and are also characterized by pain [95]. High levels of somatization were found among women with PBS [83], which correlated with increased sensitivity to experimental pain [1,85], was related to the severity of the symptoms, and predicted less therapeutic success [69,96]. It is known that psychological variables such as somatization, depression, and anxiety serve as important predictors of clinical results and the success of surgical interventions in syndromes of chronic pain [97]. However, no studies were found that examined the contribution of somatization to predicting the success of myofascial physical therapy for women suffering from CPPS.

## 5. Conclusions

In view of the above findings and due to the combination between hypersensitivity to pain and psychological factors among CPPS women, and since the pain in this syndrome affects sexual functioning and intimacy, it is important to examine the contribution of these factors, whether on their own or in interaction with physical factors, to the severity of CPPS and the success of treatment. Furthermore, the therapeutic intervention in CPPS cases should, consequently, be follow a multidisciplinary approach.

In summary, the literature indicates that CPPS represents a chronic pain syndrome that combines anatomic malfunction of the pelvic floor muscles with malfunction of pain perception linked with psychological and cognitive factors (involved in, for example, pain processing). Earlier diagnosis and support may help women to manage the syndrome and its impact on their activities of daily living. CPPS has physical, psychological, and sexual health impacts on women’s lives and those of their partners. Nurses and doctors are ideally placed to support women with CPPS. Furthermore, the therapeutic intervention in CPPS cases should, consequently, be multidisciplinary. This group of women should be investigated in RCT (Randomized Controlled Trial) studies to examine the factors that affect the syndrome and how to cope with it. 

## Figures and Tables

**Figure 1 ijerph-17-03005-f001:**
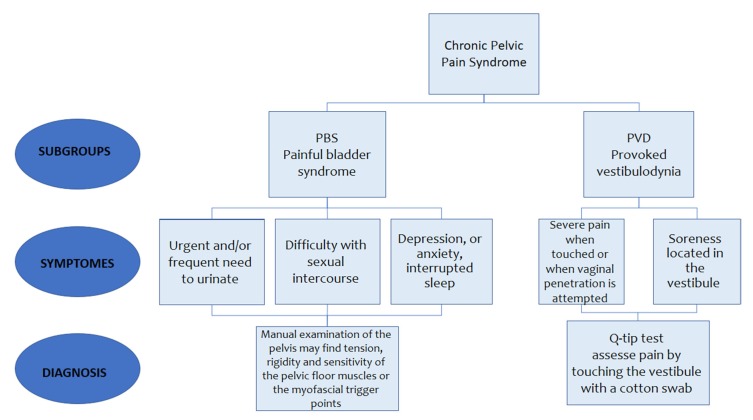
Symptoms and diagnosis of subgroups of chronic pelvic pain syndrome (CPPS).

**Figure 2 ijerph-17-03005-f002:**
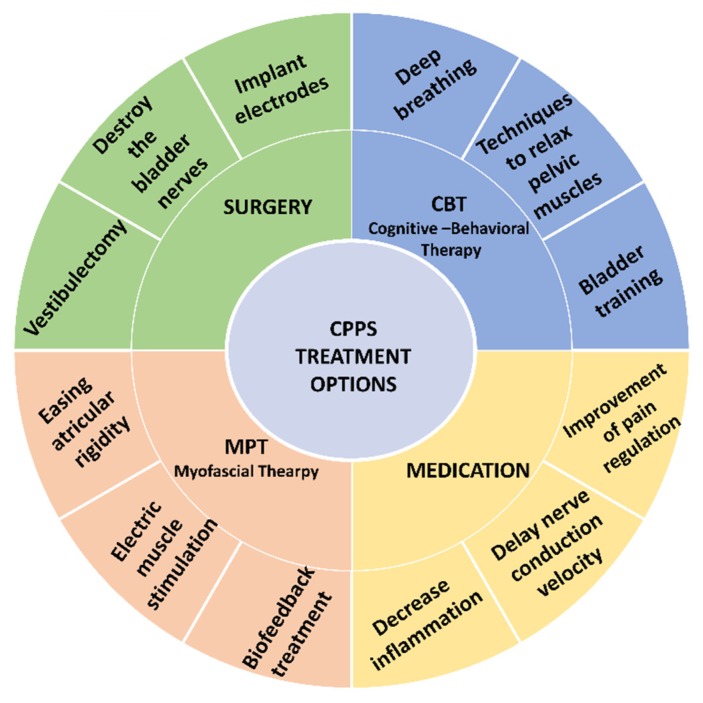
Multidisciplinary treatment options for CPPS.

**Table 1 ijerph-17-03005-t001:** Possible drug treatment interventions for CPPS.

Potential Medication	Action Mechanisms	Therapeutics Intervention
Tricyclic antidepressants; sodium pentosapolysulfate	Inhibit reuptake of noradrenaline and serotonin	Improves pain regulation; improves functioning of the downward pain-processing conduits
Resiniferatoxin; capsaicin	Bind to nerve-ending receptors and desensitize pain fibers	Delays nerve conduction velocity
Botulinum toxin	Inhibits the secretion of inflammatory agents such as substance p and nerve growth factors; inhibits production of prostaglandins	Administered due to the assumption that CPPS has an inflammatory source; prevents higher tonus of pelvic muscles
Hormone therapy	Releases hormone agonists	Some success in response to hormone therapy; pain is related to the menstrual cycle and hormonal changes

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
