# Peer review of "New Insights about Chronic Pelvic Pain Syndrome (CPPS)"

_ijerph, 2020, doi:10.3390/ijerph17093005_

Round 1

Reviewer 1 Report

Review

New insights about Chronic Pelvic Pain Syndrome

(CPPS)

The authors presented a new insight about CPPS in order to raise the awareness of the nursing and medical staff in the identification and diagnosis of the syndrome, and to promote an appropriate treatment for each women who suffer from CPPS.

This review is interesting, but should be substantial improve and corrected before publishing.

Generally points:

  1. It is necessary to improve the vividness and legibility of your manuscript. Please add to you manuscript some Figures and Tables. It can be for example a view of CPPS as a multifactorial disorder or treatment of CPPS. Or some others.. As a table you can show a short review of the literature with corresponding numbers according to your List of References.

You can decide all this things absolutely free, but you need to add some Figures and Tables to your review.   

  1. Please add to your review a section ”Materials and Methods” with search strategy, like as others review according to IJERPH. See example:

Review

Welding Fumes, a Risk Factor for Lung Diseases

Maria Grazia Riccelli 1, Matteo Goldoni 1,2 , Diana Poli 3,*, Paola Mozzoni 1,2, Delia Cavallo 3

and Massimo Corradi 1,2,4

Int. J. Environ. Res. Public Health 2020, 17, 2552; doi:10.3390/ijerph17072552

Review

Effectiveness of Educational Interventions on

Adherence to Lifestyle Modifications Among

Hypertensive Patients: An Integrative Review

Hon Lon Tam 1,2,* , Eliza Mi Ling Wong 1,* and Kin Cheung 1

Int. J. Environ. Res. Public Health 2020, 17, 2513; doi:10.3390/ijerph17072513

  1. Please describe at the end of the Introduction: how many publications and during which time period published papers were included in your review; using which data base all included paper are were found by you.

  1. What about another reviews to this topic? What about another up to date published literature? Did you include in your review the all new up to date published literature? Please include once more up to date literature to this topic.

  •  

For example: Manuelle Medizin 2019 · 57:181–187 https://doi.org/10.1007/s00337-019-0537-3 Online publiziert: 6. Mai 2019 © Springer Medizin Verlag GmbH, ein Teil von Springer Nature 2019;

Hindawi Case Reports in Urology Volume 2018, Article ID 9137215, 5 pages https://doi.org/10.1155/2018/9137215

Specially points:

Abstract

Lines 23-24: you said:  The aim of this article is to present a new insight about CPPS in order to raise the awareness of the nursing and medical staff in the identification and diagnosis of the syndrome, and to promote an appropriate treatment for each women who suffer from CPPS. You already said: Chronic pelvic pain syndrome (CPPS) is one of the common diseases in urology and gynecology.

What about CPPS in men? What about Chronic prostatitis/chronic pelvic pain syndrome?

Keywords

Please add also to keywords: Urology; Gynecology

Line 35: please add references at the end of this sentence.

Line 39: please add references at the end of this sentence.

Line 43: please add references at the end of this sentence.

Line 47: please add references at the end of this sentence.

Line 48: please add references at the end of this sentence.

Line 54:  please add references at the end of this sentence.

Line 58: please add references at the end of this sentence.

Line 61: please add references at the end of this sentence.

Line 69: please add references at the end of this sentence.

Line 73: please add references at the end of this sentence.

Line 77: please add references at the end of this sentence.

Line 90: please add references at the end of this sentence.

Line 97: please add references at the end of this sentence.

Line 136: please add references at the end of this sentence.

Line 138:  please add references at the end of this sentence.

Line 144: please add references at the end of this sentence.

Line 147: please add references at the end of this sentence.

Line 163: please add references at the end of this sentence.

Line 166: please add references at the end of this sentence.

Line 186: please add references at the end of this sentence.

Line 199:  please add references at the end of this sentence.

Line 208: please add references at the end of this sentence.

Reviewer 2 Report

This is a review paper on the topic of Chronic pelvic pain syndrome (CPPS), an important chronic multifactorial disorder which still remains difficult to dignoase and treat. The authors critically examine the available data of the literature, underlying the importance of a multidisciplinary approach to the condition.

The paper is in general well written and easy to follow, the background literature correctly reported and quoted. I only have minor concerns:

Page 2, line 39: Painful Bladder Syndrome (not Symptom) Pls correct

-Page 3, line 95: the comorbidity of CPPS with fibromyalgia is reported. I think this specific comorbidity would deserve a wider attention in the paper considering the underlying central sensitization asset. Comments on possible differences among subgroups of CPPS patients with and without fibromyalgia in terms of degree of symtpom severity and possible response to therapies would be appropriate.

-The newest IASP classification of chronic pain for ICD-11 should be quoted:

Chronic pain as a symptom or a disease: the IASP Classification of Chronic Pain for the International Classification of Diseases (ICD-11). Pain. 2019 Jan;160(1):19-27.

Author Response

The authors thanks to the reviewers for reading the manuscript and for their important comments. We have corrected the manuscript according to the comments and our answers performed below any comment (in blue color in order to make it easy to identify).

 Open Review

English language and style

( ) Extensive editing of English language and style required 
( ) Moderate English changes required 
(x) English language and style are fine/minor spell check required 
( ) I don't feel qualified to judge about the English language and style 

Comments and Suggestions for Authors

This is a review paper on the topic of Chronic pelvic pain syndrome (CPPS), an important chronic multifactorial disorder which still remains difficult to dignoase and treat. The authors critically examine the available data of the literature, underlying the importance of a multidisciplinary approach to the condition.

The paper is in general well written and easy to follow, the background literature correctly reported and quoted. I only have minor concerns:

Page 2, line 39: Painful Bladder Syndrome (not Symptom) Pls correct

Thank you for this important comment, it was corrected.

-Page 3, line 95: the comorbidity of CPPS with fibromyalgia is reported. I think this specific comorbidity would deserve a wider attention in the paper considering the underlying central sensitization asset. Comments on possible differences among subgroups of CPPS patients with and without fibromyalgia in terms of degree of symtpom severity and possible response to therapies would be appropriate.

The author reported the comorbidity of CPPS with fibromyalgia and deserve a wider attention in the paper considering the underlying central sensitization asset. Furthermore, the authors added to this topic an information from the relevant literature on the symptoms severity of in people suffering from comorbidities compared to those who did not. A little is known in the literature about the possible response to therapies among those different CPPS subgroups (with and without fibromyalgia) (row 115- 123, page 5).

-The newest IASP classification of chronic pain for ICD-11 should be quoted:

Chronic pain as a symptom or a disease: the IASP Classification of Chronic Pain for the International Classification of Diseases (ICD-11). Pain. 2019 Jan;160(1):19-27.

The authors added this important newest IASP classification to the manuscript (row 86, page 4).

Round 2

Reviewer 1 Report

Review

New insights about Chronic Pelvic Pain Syndrome (CPPS)

Thank you, this manuscript was very impressive improved.

Unfortunately, this manuscript need still little bit more corrections before publishing:

General points:

My very big problem with all Figures and Table. You said in your comments: “Thank you for your important comment. The authors added to the manuscript Figure 1, that present the symptoms and diagnosis of subgroups of CPPS (mentioned on line 65 in the revised version). The authors also added Figure 2, that describes the treatments options mentioned on line 124 in the revised version) and Table 1, that describes the possible drugs treatment interventions for CPPS (mentioned on line 145 in the revised version)”.

Unfortunately, I can not to see any Figures or Tables in your revised manuscript. Please add into the revised Version of your manuscript all Figures and Table. Please use the template of in IJERPH.

Please do it like as for example in the last Review published in IJERPH:

Review

Review: Sex-Specific Aspects in the Bariatric

Treatment of Severely Obese Women

Pia Jäger 1,2,*, Annina Wolicki 1,2, Johannes Spohnholz 1,2 and Metin Senkal 1,2

  1. Department of General and Visceral Surgery, Marien Hospital Witten, Teaching hospital of the Ruhr-University Bochum, Marienplatz 2, 58452 Witten, Germany
  2. Department of General and Visceral Surgery, Marien Hospital Herne, University hospital of the Ruhr-University Bochum, Hölkeskampring 40, 44625 Herne, Germany

* Correspondence: pia.jaeger@rub.de

Received: 18 February 2020; Accepted: 9 April 2020; Published: 15 April 2020

Special point:

Please delete the Line 45 and this sentence “Material and Methods”.

Please delete the Line 46 and the sentence “Search strategy”. Please continue after Line 44 the Line 47 with your text, without “Materials and Methods” section. I think, thus it is much better.  Now you described all this things about the literature searching very well in your Introduction section.

Line 59: please add the references at the end of this sentence.

Line 68: please add the references at the end of this sentence.

Line 70: please add the references at the end of this sentence, after “cotton swab”.

Lines 74-76: please add the references at the end of this sentence after 1) and 2).

Lines 76-80: please correct this sentence. It is something wrong with this sentence.

Line 81: please add the references at the end of this sentence.

Lines 130-132: please correct this sentence. It is something wrong with this sentence.

Line 175: please add the references at the end of this sentence.

Line 177: please add the references at the end of this sentence.

Line 188: please add the references at the end of this sentence.

Line 243: please add the references at the end of this sentence.

Line 246: please add the references at the end of this sentence.

Line 280: please correct this sentence. It is something wrong with this sentence.

Line 287: please correct this sentence. It is something wrong with this sentence.

Author Response

The authors thanks to the reviewers for reading the manuscript and for their important comments. We have corrected the manuscript according to the comments and our answers performed below any comment (in blue color in order to make it easy to identify).

Open Review

 ( ) Extensive editing of English language and style required 
( ) Moderate English changes required 
(x) English language and style are fine/minor spell check required 
( ) I don't feel qualified to judge about the English language and style 

Comments and Suggestions for Authors

Review

New insights about Chronic Pelvic Pain Syndrome

(CPPS)

The authors presented a new insight about CPPS in order to raise the awareness of the nursing and medical staff in the identification and diagnosis of the syndrome, and to promote an appropriate treatment for each women who suffer from CPPS.

This review is interesting, but should be substantial improve and corrected before publishing.

Generally points:

  1. It is necessary to improve the vividness and legibility of your manuscript. Please add to you manuscript some Figures and Tables. It can be for example a view of CPPS as a multifactorial disorder or treatment of CPPS. Or some others.. As a table you can show a short review of the literature with corresponding numbers according to your List of References.

You can decide all this things absolutely free, but you need to add some Figures and Tables to your review.

Thank you for your important comment. The authors added to the manuscript Figure 1, that present the symptoms and diagnosis of subgroups of CPPS (mentioned on line 65 in the revised version). The authors also added Figure 2, that describes the treatments options mentioned on line 123 in the revised version) and Table 1, that describes the possible drugs treatment interventions for CPPS (mentioned on line 144 in the revised version).

  1. Please add to your review a section ”Materials and Methods” with search strategy, like as others review according to IJERPH. See example:

Review

Welding Fumes, a Risk Factor for Lung Diseases

Maria Grazia Riccelli 1, Matteo Goldoni 1,2 , Diana Poli 3,*, Paola Mozzoni 1,2, Delia Cavallo 3

and Massimo Corradi 1,2,4

Int. J. Environ. Res. Public Health 2020, 17, 2552; doi:10.3390/ijerph17072552

Review

Effectiveness of Educational Interventions on

Adherence to Lifestyle Modifications Among

Hypertensive Patients: An Integrative Review

Hon Lon Tam 1,2,* , Eliza Mi Ling Wong 1,* and Kin Cheung 1

Int. J. Environ. Res. Public Health 2020, 17, 2513; doi:10.3390/ijerph17072513

The authors added to the manuscript, a new "Materials and Methods" section with search strategy (row 42, page 2).

  1. Please describe at the end of the Introduction: how many publications and during which time period published papers were included in your review; using which data base all included paper are were found by you.

The authors added this details to the end of the "Methods and Materials" section (row 47, page 2). If the reviewer think that it might be at the end of the Introduction section, it will be corrected as requested to the end of the introduction section.

  1. What about another reviews to this topic? What about another up to date published literature? Did you include in your review the all new up to date published literature? Please include once more up to date literature to this topic.

For example: Manuelle Medizin 2019 · 57:181–187 https://doi.org/10.1007/s00337-019-0537-3 Online publiziert: 6. Mai 2019 © Springer Medizin Verlag GmbH, ein Teil von Springer Nature 2019;

Hindawi Case Reports in Urology Volume 2018, Article ID 9137215, 5 pages https://doi.org/10.1155/2018/9137215

Thank you for the comment and the examples. The authors included once more up to date literature to this topic (row 36 page 2, row 122 page 5).

Specially points:

Abstract

Lines 23-24: you said:  The aim of this article is to present a new insight about CPPS in order to raise the awareness of the nursing and medical staff in the identification and diagnosis of the syndrome, and to promote an appropriate treatment for each women who suffer from CPPS. You already said: Chronic pelvic pain syndrome (CPPS) is one of the common diseases in urology and gynecology.

What about CPPS in men? What about Chronic prostatitis/chronic pelvic pain syndrome?

CPPS is also existing among men and presented as chronic prostatitis and related to the Urology domain. This review aimed to score the CPPS of women which related to the Urology and gynecology domain in order to present the other subgroups of the disease (such as PVD) that related only to women and reflected as a chronic pelvic pain syndrome under some different pathogenesis, (such as inflammation of the prostate). Some women suffer from an urological symptoms- Painful Bladder Syndrome (PBS), because of that it is also common diseases in urology and gynecology.

If the reviewer still thinks we should delete the name women and / or urology the authors will correct it.  

Keywords

Please add also to keywords: Urology; Gynecology

 The authors added to the end of the abstract key words: Urology; Gynecology (row 32 page 1)

Line 35: please add references at the end of this sentence.

The authors added references (in the revised version -line 37)

Line 39: please add references at the end of this sentence.

There are references (in the revised version -line 59)

Line 43: please add references at the end of this sentence.

The authors added references (in the revised version - line 62)

Line 47: please add references at the end of this sentence.

The authors added references (in the revised version - line 64)

Line 48: please add references at the end of this sentence.

The authors added references (in the revised version - line 67)

Line 54:  please add references at the end of this sentence.

The authors added references (in the revised version - line 73-74)

Line 58: please add references at the end of this sentence.

The authors added references (in the revised version - line 81)

Line 61: please add references at the end of this sentence.

The authors added references (in the revised version - line 84)

Line 69: please add references at the end of this sentence.

The authors added references (in the revised version - line 85-86)

Line 73: please add references at the end of this sentence.

The references is presented in the beginning of this sentence, line 89-90 in the revised manuscript

Line 77: please add references at the end of this sentence.

The references is presented in the beginning of this sentence, line 92 in the revised manuscript

Line 90: please add references at the end of this sentence.

The authors added references (in the revised version - line 95-97)

Line 97: please add references at the end of this sentence.

The authors added references (in the revised version - line 103-104)

Line 136: please add references at the end of this sentence.

The authors added references (in the revised version - line 155-157)

Line 138:  please add references at the end of this sentence.

This is the end of the sentence in line 136, the authors added references (in the revised version line 160)

Line 144: please add references at the end of this sentence.

The authors added references (in the revised version - line 168)

Line 147: please add references at the end of this sentence.

The authors added references (in the revised version - line 170-171)

Line 163: please add references at the end of this sentence.

The authors added references (in the revised version - line 172)

Line 166: please add references at the end of this sentence.

The authors added references (in the revised version - line 178-179)

Line 186: please add references at the end of this sentence.

The authors added references (in the revised version - line 208-211)

Line 199:  please add references at the end of this sentence.

The authors added references (in the revised version - line 221)

Line 208: please add references at the end of this sentence.

The authors added references (in the revised version - line 230)

Round 3

Reviewer 1 Report

Thank you for multiple corrections. This manuscript was impressivly improved according all my suggestions.

I have only two small further proposals:

Please correct the line 256.

Please check and correct the distance between the Table 1. and line 149.

Author Response

Thank you for your comments. We updated this version regarding to your request and have added a conclusion section, lines 241-246.